# Development and Characterization of Chitosan and Porphyran Based Composite Edible Films Containing Ginger Essential Oil

**DOI:** 10.3390/polym14091782

**Published:** 2022-04-27

**Authors:** Ahmed Al-Harrasi, Saurabh Bhtaia, Mohammed Said Al-Azri, Hafiz A. Makeen, Mohammed Albratty, Hassan A. Alhazmi, Syam Mohan, Ajay Sharma, Tapan Behl

**Affiliations:** 1Natural & Medical Sciences Research Center, University of Nizwa, P.O. Box 33, Birkat Al Mauz, Nizwa 616, Oman; malazri@unizwa.edu.om; 2School of Health Science, University of Petroleum and Energy Studies, Dehradun 248007, India; syammohanm@yahoo.com; 3Pharmacy Practice Research Unit, Clinical Pharmacy Department, College of Pharmacy, Jazan University, Jazan 45142, Saudi Arabia; hafiz@jazanu.edu.sa; 4Department of Pharmaceutical Chemistry, College of Pharmacy, Jazan University, P.O. Box 114, Jazan 45142, Saudi Arabia; malbratty@jazanu.edu.sa (M.A.); hasalhazmi@gmail.com (H.A.A.); 5Substance Abuse and Toxicology Research Center, Jazan University, Jazan 45142, Saudi Arabia; 6Department of Pharmacognosy & Phytochemistry, School of Pharmaceutical Sciences, Delhi Pharmaceutical Sciences and Research University, New Delhi 110017, India; ajaysharmapharma1979@gmail.com; 7Chitkara College of Pharmacy, Chitkara University, Rajpura 140401, India; tapanbehl31@gmail.com

**Keywords:** edible films, composites, chitosan, porphyran, ginger essential oil, glycerol, food packaging, antioxidant

## Abstract

Recent research shows the growing interest in the development of composite edible films (EFs) by using multiple biopolymers for the substantial improvement in the shelf life and quality of food products, via preventing oxidation among other benefits. In the present work, EFs based on chitosan (CS) and porphyran (POR) loaded with ginger essential oil (GEO) have been developed to study the effect of GEO, glycerol (Gly), and POR on the film structure as well as physical and antioxidant properties. Fourier transform infrared spectroscopy (FTIR) and X-ray diffraction (XRD) results showed the level of crystallinity and electrostatic interactions between CS, POR, Gly, and GEO. It was found that electrostatic interactions between CS and POR and the incorporation of GEO substantially improved barrier, thermal, optical, and mechanical properties and reduced the moisture content, swelling index, and thickness values. The color values of the S5 film altered apparently with a shift towards yellowness. SEM micrographs of the composite CS-POR-GEO film (S5) showed improved morphological attributes such as more uniformity and homogeneous structure than other films (S1–S4). Results obtained from total phenolic content assay suggested the presence of high phenolic components (5.97 ± 0.01) mg of GAE/g in GEO. Further, findings obtained from antioxidant assays revealed that the addition of GEO and POR significantly increased the antioxidant effects of CS films. All these findings suggested that GEO loaded CS-POR based films showed better physical and chemical properties with a significant improvement in antioxidant potential and thus can be used as a potential packaging material in the food industry.

## 1. Introduction

To overcome the negative environmental impact of petroleum-based packaging, recently, natural polymer based edible films (EFs) with biodegradable, biocompatible, and other potential features have been considered as their smart replacements. These EFs have been recently reported as carriers for integrating an extensive range of additives, including natural antioxidants as well as antimicrobials. Moreover, these bioactive natural coating materials have been reported for their effect in diminishing unfavorable reactions (enzymatic, physical, and chemical reactions) and improving surface, thermodynamic, mechanical, and barrier properties [1]. Owing to growing concerns related with food quality and safety, one of the most attractive ways adopted by various researchers is based on the fabrication of EFs loaded with natural antioxidant agents to enhance shelf life as well as maintain food quality via reducing food oxidation. In comparison to lipid- as well as protein-based films, natural polysaccharide based edible coatings have the benefits of cost effectiveness, ample resources, and comparatively more stability with improved thermodynamic and water solubility properties. Chitosan is a cationic polysaccharide that has been reported as safe, biodegradable, edible and non-toxic polysaccharide known for its capability in forming transparent films with good mechanical properties. However weak vapor permeability as well as hydrophilic character have restricted its utilization as potential coating material. Recent research has shown that the development of chitosan based composite coating material can overcome such challenges. Porphyran is an anionic low molecular weight sulphated algal polysaccharide with high antimicrobial, antioxidant, anticancer and immunomodulatory effects [2,3]. The incorporation of an anionic polysaccharide such as POR can improve the physicochemical properties of chitosan-based films as it can form stable PEC with improved water resistance properties (Figure 1) [2,3]. As studied in our previous work, CS and POR form a stable stoichiometric complex via forming a strong electrostatically bond between amine group of CS and sulphate group of POR, resulting in improved properties of CS [2,3]. Figure 1 shows a possible interaction between CS and POR to form a polyelectrolyte complex (PEC) complex in the presence of GEO components, glycerol, and tween 80. Additionally, to improve the water-resistant properties of CS, besides adding lipid or any fatty material, natural aromatic, antimicrobial, and antioxidant agents in the form of essential oils (EO) can be incorporated alternatively. This approach can not only improve the water-resistant property of CS, but can also avoid the diffusion of EO into packed material. Along with EO, glycerol addition as a plasticizer to the composite material assists in decreasing the cohesive forces among polymeric matrix as well as improving the fluidity of the polymer chain. This addition of EO with plasticizer can further decrease film brittleness and improve the film’s flexibility.

Ginger essential oil (GEO) has been reported for its strong antioxidant effects [4]. This antioxidant activity could be associated with the presence of phenolic compounds in GEO. On another hand, CS and POR have been also reported for their potential antioxidant effects [2,5]. Thus, both CS and GEO can act synergistically against free radicals in food components to prevent any chemical changes in the food. Thus, the objective of the present work was to fabricate blank CS, plasticized CS, and GEO loaded plasticized CS EFs, and to compare their physio-chemical and antioxidant properties with blank plasticized CS-POR and GEO incorporated CS-POR films.

## 2. Results and Discussion

### 2.1. Visual Appearance of the EFs

Visual characterization of the coating material meant for food packaging is important to assess its physical attributes, e.g., color, transparency, mechanical strength, brittleness, fragile, sticky as well as rigid nature, flexibility, difficulty or easiness during peeling, etc. This characterization usually helps in the initial screening of the films. Figure 2 and Table 1 present attributes based on visual screening. CS films (S1 and S2) prepared with and without glycerol showed a fragile, shiny, brittle, stiff, less flexible appearance with several cracks over the surface of the EFs.

Owing to fragile, non-flexible, and brittle nature, S1 film (CS blank) faced more difficulty than S2 while peeling them from the surface of the Petri plates. This could be due to the compact and dense structure formed in S1 due to strong inter/intra molecular hydrogen bonding of CS that restricted movement and flexibility of the polymeric chains, leading to formation of more fragile and brittle EFs. As shown in Figure 2 (side view), peeling S1 disintegrated into the pieces, which makes the peeling process more difficult as more force was required to peel it off from the Petri plate surface. S2 (CS with glycerol), due to plasticization effect induced by glycerol, showed a decrease in intermolecular hydrogen bonding and rise in molecular mobility. This behavior could be due to the role of glycerol in restructuring intermolecular polymer chain networks as its lower molecular weight allows its entry into intermolecular spaces of polymer matrices. Similar behaviors of the films were reported in the previous work [6,7]. CS films with plasticizer and GEOs (S3) possessed a rigid, fragile, brittle, rough surface, with less transparency and flexibility than S1 and S2. Like S1, S3 also faced challenge during peeling, perhaps due to the interference to the solubility of glycerol caused by GEO. CS-POR films (S4, S5) were not rigid, fragile, or brittle, showing more flexibility and smoothness, and were easily peeled from the Petri plate surface and showed less transparency than S1–S3. CS-POR-GEO EFs (S5) with plasticizer demonstrated higher flexibility, followed by CS-POR-plasticized films as the addition of GEO resulted in the weakening of intermolecular hydrogen bonds of CS-POR films. The molecular interaction between CS and POR by forming stable PEC could be attributed to improved properties of the film which do not allow the GEO to interfere with solubility of glycerol. Thus, the addition of Gly, POR, and GEO made the peeling of films easier without any disintegration resulting in overcoming crumbliness offered by CS film via softening their 3-D molecular structural configuration and increasing the molecular free volume. The molecular interaction between plasticizer (Gly) and CS-POR decreased the cohesive strain of the CS molecules. Furthermore, despite their merits, the appearance of all the CS-POR was less transparent with a slight yellowish appearance, which is positive on one side as this color shift can prevent food degradation caused by UV light and negative on the other side as it can impact the consumer acceptability. Color shift can be overcome by optimization of the ingredients in future.

### 2.2. Film Thickness

The composite film thickness varied in the range of 46.2–58.1 μm (Table 2), and the incorporation of GEO significantly (*p* > 0.05) affected the resulting film thickness. Thickness of the CS-POR (S4, S5) films was found to a lesser extent than in the CS films (S1–S3). This is due to the formation of PEC between CS-POR, which can enhance the interaction between the molecules and reduce the number of voids, thereby reducing the thickness of the films. On other hand, S3 showed more thickness than S1 and S2, and similarly S5 showed more thickness than S4. This behavior was because of the incorporation of GEO. As suggested in previous reports, an increase in thickness could be due to the interaction between the film and GEO or it could be due to the formation of micro-droplets from the hydrophobic GEO [8].

### 2.3. Mechanical Properties

Ideally, a food packaging material must have sufficient mechanical properties, structural integrity, and better flexibility to mainly tolerate external stress and to maintain its barrier properties during packaging application. Mechanical properties in this study were determined by measuring Y (Young modulus), TS (tensile strength), and EB (elongation at break). Usually, the chemical composition of the EFs determines mechanical properties of the film. TS suggests the resistance against tension forces, EB is stretching capacity of the edible films or the material material’s deformation capacity, and Y can be explained as a measurement of the stiffness or the rigidity of EFs. Findings relating to Y, TS, and EB are presented in Table 2. Young’s modulus values of the CS based films (S1–S3) were significantly higher than those of CS-POR (S4, S5). S4 and S5 samples showed higher tensile strength than S1–S3. The increasing TS values of the composite films (S4 and S5) are attributable to the PEC formation resulting in intermolecular hydrogen bonds formation between NH_2_ of the chitosan backbone and exposed OH^−^ as well as the sulphur group of the POR. The TS of GEO loaded samples (S3, S5) was found to be relatively lesser than S1, S2, and S4 because of the incorporation of GEO. The decrease in TS could primarily be attributed to the GEO, as GEO dispersion into polymeric networks decreased the molecular interactions (inter- and intra-) which ultimately diminished the continuous film matrix. This can further lead to domination of weaker polymer-GEO interactions over stronger intermolecular polymer interactions in EFs matrix. Similar findings have been observed when lavender EO was added into the gelatin based ternary potato starch film [9]. EB values of the S4 and S5 were found to be higher than that of S1–S3. It was found that elongation as well as elasticity of the S1–S3 EFs was enhanced with the incorporation of the porphyran. S4 and S5 samples showed an improvement in elongation as well as elasticity, suggesting that this fabricated material can be utilized for future food packaging application [9].

In addition, GEO loaded samples (S3, S5) showed higher EB values than others because of the strong plasticizing effect possessed by the incorporated GEO. This is due to the enhancement in the promoted mobility of polymeric chains resulting in further improvement in the flexibility of the film, by this means improving the EB of the EFs [8]. It was found that the incorporation of *Zataria multiflora* EO also augmented the EB of the gelatin film [10]. Similar findings were noted in our work. Effect of glycerol addition on mechanical properties of CS (S2) sample has been observed as it significantly decreased Y and TS (*p* < 0.05) and increased EB when compared to CS blank (S1). The glycerol addition reduced the rigidness and brittleness of unplasticized CS blank (S1) films.

### 2.4. Water Vapor Permeability (WVP)

WVP is a crucial feature of EFs fabricated as a food packaging material. Ideally WVP values of edible films or any food packaging material must be as minimum as possible to reduce water transmission between the environment and food to avoid food spoilage in order to keep the packed food fresh always. WVP values of samples (S1–S5) are demonstrated in Table 2. WVP of the CS based films was found to be higher than CS-POR films. This behavior could be due to the PEC formation between two polymers (CS-POR) because the decrease in the mobility of the polymeric chains resulted in the formation of denser and stable polymeric networks. Moreover, it was found that CS blank films (S1) showed higher WVP than GEO loaded CS EFs (S3). This could be due to the presence of Gly in S2 due to which high hydrophilicity of the glycerol molecule favored the adsorption of water molecules and could lead to rise in the film WVP. Further, the addition of glycerol increases inter chain spacing along the polymeric chain, which may increase WVP across the S1. However, the loading of GEOs in CS or CS-POR plasticized films (S3 and S5) resulted in the drop in WVP because of the nonpolar nature of GEOs. As mentioned in Table 2, WVP of S3 was found to be higher than S5 due to interference caused by GEO on the microstructure of films, resulting in the rise in cracks or pores in the EF structure, leading to flow of water vapor molecules. These findings can also be supplemented with the SEM images. Previous research work demonstrated structural modification caused by the GEO components in the polymeric network, by decreasing the density and increasing the WV transmission across films [11]. However stable PEC formed by CS-POR (S5) has not allowed GEO to interfere with properties of PEC apart from contributing hydrophobicity, resulted in a decrease in WVP.

### 2.5. Oxygen Barrier Properties

Food oxidation due to the availability of oxygen causes changes in food, which can significantly impact the shelf life of packed material. Thus, preservation of the food is also dependent on the oxygen permeability (OP) of films. Food preservatives especially available in the market reduce the oxygen level to prevent spoilage of the food. These oxygen absorbers have the potential to reduce the dissolved oxygen, and thus maintain quality of the food by preventing its spoilage during storage.

Usually, in comparison to others, hydrophilic films exhibit excellent oxygen barrier properties. Table 2, representing the oxygen permeability of EFs (S1–S5), shows that incorporation of GEO and POR decreased OP. This could be due to the PEC formation resulting in a complex and compact construct, which may restrict the movement of oxygen molecules (non-polar) across the film. Moreover, the antioxidant components of GEO can reduce the free radicals present in the food, resulting in an improved OP property of the EFs. CS-POR composites showed a decrease in OP values and this improvement in O_2_ barrier properties might be due to the PEC formation of intermolecular bonding between functional groups of GEO and polymers resulting in the formation of dense and compact structure with low OP value. It was also reported that the incorporation of GEO caused a substantial drop in OP due to its hydrophobic nature [12]. S1–S3 samples showed greater OP values than S4 and S5 samples. This behavior could be due to the presence of more pores and cracks over the surface and cross section (as shown in SEM micrographs, Figure 5). Additionally, in the S2 sample (in comparison to S1), the addition of glycerol caused decrease in the OP value. This tendency of the EF could be due to an increase in the movement of polymer chains inside the film matrix to restrict the entry of oxygen through the film.

### 2.6. Water Solubility (WS) and Swelling Degree (SD)

Water solubility is an important parameter as it determines the water resistance offered by the edible films having food packaging applications, mainly in an environment with high humidity. Water susceptibility is one of the main drawbacks of films made using biopolymers. Water retention or resistance of these polymers is significantly influenced by the hydrophilic or hydrophobic nature of the ingredients added to the formulation. Table 3 shows the water solubility from S1 to S5 formulations. Chitosan is water insoluble whereas porphyran is a water-soluble sulphated polysaccharide. Molecular interaction between two oppositely charged polymers (CS-POR) leads to the formation of PEC by forming more hydrogen bonds, thus causing a decrease an availability of free –OH over the surface of the films. Consequently, due to this water solubility, S1 was found to be greater than S4. Moreover CS-Gly films showed more water solubility than CS alone. This behavior is due to the high hygroscopic nature of glycerol, which can considerably improve the moisture content of edible films. This can overcome brittleness of the film however make the film more hydrophilic [13]. As shown in Table 3, WS of the samples considerably decreased (*p* < 0.05) with the incorporation of GEO. Findings demonstrated that the incorporation of GEO enhanced the water insolubility or resistance of the films. This could be due to the reduction in the hydrophilic nature of the films after addition of the hydrophobic EO. Additionally, components of GEO can also interfere with the interaction of hydroxyl groups of the polymers and plasticizer with water molecules, resulting in the development of more water resistance [14].

In contrast to the swelling degree of the films, the incorporation of GEO and POR led to the reduction in the swelling index (*p* < 0.05). This could be due to the interference caused by the GEO and POR with the effect of a plasticizer to absorb water and cross linking between CS and POR. This molecular interaction between CS-POR decreases the capacity of the EFs to retain water. This type of interaction between two oppositely charged ions prevents the chitosan functional groups to interact with the water and hence weaken its ability to absorb it [14].

### 2.7. Moisture Content

Films meant for food packaging applications must be ideally water resistant. One of the major drawbacks of EFs obtained from natural polymers is their capability to absorb water vapor present in the environment. Usually, the hydrophilic or hydrophobic nature of the film components determines the moisture retention potential of film. Table 3 shows the moisture content (MC) of the prepared films. The MC of plasticized CS film (S2) was found to be more than CS blank film (S1) as this hygroscopic plasticizer has the tendency to retain moisture content to prevent the brittleness of the film. Thus, Gly addition significantly increased the MC values in CS plasticized films, which could be associated with its hygroscopic nature [13]. However, the incorporation of GEO reduced MC content because of the rise in hydrophobicity level of the S3 and S5 EFs. This could be due to the electrostatic interaction among the chemical moieties of polymers (CS and POR) and GEO resulting in the reduction in the accessibility of amino as well as hydroxyl moieties and decrease in hydrogen bonding among the polymers and the water molecules [2]. These findings could be due to the good emulsifying action of POR and Tween 80 that enhanced the GEO dispersion in the polymeric matrix [1], resulting in a decrease in the ability of films to bind water molecules [2]. Further, surprisingly, the plasticized CS-POR composite without GEO (S4) showed a lower MC value than S2. This behavior could be due to PEC formation, which may reduce the sensitivity of glycerol against water or decrease the availability of free –OH over the surface of the films.

### 2.8. Optical Transmittance Analysis

Optical transmittance related attributes, such as color and transparency, are important parameters that clearly influence the end user acceptability and film appearance. Transparency is a crucial parameter used to assess transparency of the film where greater transparency value suggests lower film transparency. Transparency of the films (S1–S5) was displayed in Table 4. The light transmittance of the CS blank film (S1) was found to be the highest (42.8%) among all films. Optical transmission from S1 to S4 reduced considerably (*p* < 0.05) due to the addition of plasticizer and GEO. Moreover, the formation of PEC between CS-POR (S4, S5) resulted in the development of more compact films with less porosity and no cracks (as demonstrated in SEM findings). Thus, S4 and S5 films showed less light transmittance. The addition of Gly and GEO resulted in the formation of films carrying oil or water droplets which can interfere with light transmission and cause more scattering of the light at the boundary of drops. These findings demonstrated that GEO, Gly, and POR could efficiently prevent the transmission of ultraviolet radiation from the prepared EFs (S5). Moreover, due to the presence of colorful components of GEO, films loaded with GEO showed less transparency. From commercial aspects, these composite films loaded with GEOs can prevent UV radiation transmission and thus offer more protection to packed, light-sensitive food from oxidation and other extrinsic food damaging sources [15].

### 2.9. Color Parameters of the EFs

Film color is an important attribute that determines the consumer suitability based on the appearance of the packaged products. As per previous reports, incorporation of EO in EFs significantly impacts its appearance, such as the transparency as well as color of EFs [16]. Color measurements of the EFs loaded with and without GEO are shown in Table 4, and the form of the EFs is demonstrated in Figure 1 and Table 1. Control films (S1) showed cracks, namely tiny particles over the surface, whereas S2 (CS with plasticizer) showed a relatively rougher surface. It was found that after the incorporation of GEO to S1, CS films (S3) showed significant (*p* < 0.05) changes in terms of color and transparency of the films. As can be noticed from Figure 1, the color characteristics of S3 and S5 clearly shifted from transparent to pale yellow. As demonstrated in Table 4, the yellow color of GEO considerably contributed to the yellowness (b*) of the S3 and S5 films. This finding was consistent with the earlier finding where the incorporation of cinnamon EO to the Sago starch films impacts its color characteristics [16]. The addition of GEO caused a decrease in lightness (L*), with a rise in total chromatic difference (ΔE*), greenness (a*), and chromatic intensity (CI*) values of the GEO-loaded EFs. Darkening of the films could be due to the light scattering in the GEO droplets. This scattering of light inside the droplet caused diffuse reflectance, resulting in a decrease in the whiteness index and light scattering intensity of the films. These findings showed that the addition of GEO can change the film color properties by yellowing as well as darkening in color due to the colorful component present in the GEO. Similar trends were observed in the previous reports.

### 2.10. TGA Analysis

The thermal stability of the prepared EFs (S1–S5) were measured by TGA to study the impact of POR, Gly, and GEO over the thermal properties of the EFs. Findings obtained (S1–S5) revealed various stages of weight variation in the TG curve, as shown in Figure 3. The first weight loss was evidenced in the range of 57–130 °C, which might be ascribed to the loss of free or bound water. All prepared EFs showed the first weight loss at the temperature of 57 °C. The next phase of weight loss was observed in the range of 135–398 °C which was attributed to the hydrolysis of POR, volatilization of GEO, Gly, and Tween 80, as well as degradation due to the disaggregation of CS-POR complex [2]. Subsequently, weight variation in the temperature range of 412–600 °C was due to the degradation of the polymers in the EFs. However, in terms of the film containing only chitosan, especially S1 and S2, a new peak was observed at approximately 260 °C, terminating at 340 °C, and this could be due to the decomposition of chitosan which was accountable for weight loss [17]. TGA study demonstrated that the incorporation of the GEO and POR enhanced the thermal stability of the EFs. This property was perhaps due to the interaction among the two oppositely charged polymers (CS and POR) to form a more stable PEC (CS-POR) system [2]. Therefore, these EFs can be considered as suitable food packaging materials even when used at a comparatively elevated temperature.

### 2.11. X-ray Diffraction (XRD) Analysis

XRD patterns of test samples are usually assessed to analyze the internal atomic space structure of a substance by using the diffraction impact of X-rays in the crystal structure. XRD was performed to study the compatibility of CS and POR with Gly and GEO by assessing the amorphous nature or crystallinity of the fabricated films (S1–S5). Figure 4 shows the XRD spectral patterns of CS-POR based films. The XRD spectral peaks of EFs demonstrated well-defined characteristic peaks. All edible films with different composition showed similar diffraction patterns in a range of 14.5–25.9°, indicating that most of the structure in the EFs produced was amorphous. The XRD patterns noted showed good compatibility among CS, POR, Gly, and components of GEO as the peaks were broader. A broader XRD peak also suggests a partial or less crystalline structure. An earlier report suggested a similar diffraction peak (20°), signifying the amorphous nature of EFs synthesized from carrageenan and locust bean gum [18].

The crystallinity of the CS-POR based films (S4, S5) was significantly lower than that of the CS based films (S1–S3). Furthermore, the crystallinity of the EFs with GEO was considerably less than that of the EFs without GEO under similar experimentation conditions. This suggested that the incorporation of GEO may decrease the CS-POR crystallinity. The decrease in the intensity of the peak could be associated with the intermolecular interaction between CS-POR, resulting in a reduction in the mobility of molecules and thus preventing crystallization.

### 2.12. SEM

Microstructural characteristics of CS control (S1), CS-Gly (S2), CS-GEO-Gly (S3), CS-POR-Gly (S4), and CS-POR-Gly-GEO (S5) films at different magnifications are presented in Figure 5. Microstructural properties of these samples were examined to study the combined effect of glycerol, GEO, and POR over the properties of CS blank film. Surface characteristics of CS (control) (S1) using CS with 90% DA was observed as rough, with cracks, pores, and tiny particles over the surface (Figure 5). This was contradictory to the results obtained from previous investigation [7]. SEM characteristics of CS films with plasticizer (S2) showed bulges, a rough surface, and tiny particles over the surface with fewer cracks (Figure 5). GEO loaded CS films with plasticizer (S3) presented similar surface characteristics as like chitosan blank. This could be due to the interference caused by GEO over the plasticization effect of Gly and film forming ability of chitosan. Although CS-POR (S4) and CS-POR-GEO (S5) films with Gly showed no cracks, bulges, roughness and especially CS-POR-EO films showed a smoother surface with less appearance of tiny particles over the surface than other films (S1–S4) (Figure 5). Cross section studies of CS control (S1), CS-Gly (S2), CS-Gly-GEO (S3), CS-POR-Gly (S4), S5, and S4 samples revealed no pores with more regular, dense, and compact structure than S1–S3. S1–S3 samples showed discontinuity in the polymeric structure with a slight appearance of porous structures that could be due to the volatilization of the GEO in S3.

### 2.13. FTIR Analysis

During the synthesis of EFs, the incorporation of different components in polymeric matrices and its respective chemical interaction are suggested by shift in their typical absorption bands. FTIR spectra of S1–S5 samples demonstrated in Figure 6 show molecular interactions between different CS, POR, GEO, and Gly. All samples (S1–S5) demonstrated quite similar spectra, with most of the peak characteristic of CS films. In the spectra of all films, the broad peaks in between 3000 and 3500 cm^−1^ were caused by O–H stretching vibration. These peaks became more flattened in S4 film spectra because of the presence of POR. However, this peak was again retained after addition of GEO in S5 film. CS films (S1–S3) showed more intense peak in the region from 3000 nm to 3500 cm^−1^ than CS-POR films (S4 and S5), which might be ascribed to the greater interaction of chemical moieties in GEO with CS (–OH or –NH_2_) and POR (SO_4_ or –OH) of molecules, or more interaction between functional groups of CS and POR causing decrease in N–H and O–H stretching [19]. Peaks at 1155 and 821 cm^−1^ are related to S=O bond stretching and attachment of the sulfate group to the primary hydroxyl group. The absorption band at 927 cm^−1^ is related to 3,6-anhydrogalactose residues in the polysaccharide chain. The existence of 3,6-anhydrogalactose and sulfur residues proves that the sample is sulphated polysaccharide (i.e., POR) [2].

The absorption bands appeared between 993 and 1153 cm^−1^ linked to the stretching of the C-O bond. While formulating films, the mixing of various ingredients facilitates chemical interactions, which is represented by shifting in characteristic bands. In the spectra of S4 and S5 films, the amino peak of CS shifted from 1560 cm^−1^ to 1598 cm^−1^ with the addition of POR. Moreover, shifting of the sharp peak of S1 at 1037 cm^−1^ to 1055 cm^−1^ in S4 and 1056 cm^−1^ in S5 along with the complete disappearance of 1110 cm^−1^ peak of S1 sample showed interactions between the chemical moieties of GEO, CS, and POR. The characteristic peaks due to the presence of carboxyl group at 2923 cm^−1^, the aromatic ring (C=C) at 1581 cm^−1^, the alkene CH group at 1602 cm^−1^, and peaks at 1624, 1265, and 621 cm^−1^ were related to the C–O as well as C=C stretching and H-atom in the aromatic ring, demonstrating the existence of phenolic groups in the S3 as well as S5 GEO loaded films. Generally, the addition of GEO did not cause major changes in the spectra perhaps due to the addition of little amount. Thus, the characteristic peaks of CS existed among all film samples. Nevertheless, slight variations in the intensities of the absorption bands were documented, which are ascribed to the similarity of chemical bonds suggesting good interaction between the functional groups of the various elements of the EFs. The appearance of new peaks in S4 at 898 and 937 cm^−1^ or 1218 cm^−1^ and the shifting of peaks at different wavelengths with the addition of GEO suggested the formation of new covalent bonds between the CS-POR and active components of GEO. These absorption spectra advocated possible interactions that developed between CS-POR complex and GEO components via covalent and hydrogen bonding, facilitating the formation of a more compact and homogeneous film structure.

### 2.14. Antioxidant Assays

The oxidation of food is a deteriorating reaction that ultimately causes changes in its chemical, sensory and nutritional properties. Synthetic antioxidants, such as butylated hydroxyanisole, butylated hydroxytoluene, and many others, have been recently criticized for their safety concerns. The recent trend in utilizing natural antioxidants loaded EFs has overcome challenges associated with possible toxic effects caused by synthetic antioxidants. The composition of the edible films usually determines its antioxidant potential. Incorporation of natural antioxidants can retard the oxidation of proteins, lipids, and other free radicals, which are sensitive components of the food. This can result in a drastic improvement in the quality, stability, and overall shelf life of food products. Moreover, adding natural antioxidants can preserve or enhance the sensory properties of the food. Growing interest in using EO as a natural antioxidant especially via loading them in edible films has shown more promising results in improving the shelf life, stability, as well as sensory properties of packed food. There is always a chance of leaching of essential oils from film to inside the food material which can influence its chemical composition as well as the sensory properties. However, the leaching of essential oil components from the film to the packed food material for relatively shorter duration would have a less negative impact than the leaching of harmful components from the plastic material for longer duration. Moreover, less concertation, non-polar and volatile nature would not allow its entry inside to the food material Thus, essential oil components act on the surface to improve their shelf life and impart the characteristic odor and flavor to the packed food material. Additionally, the incorporation of EO can provide antioxidant and antimicrobial effects to the films. Most EOs have been categorized as GRAS (generally recognized as safe), at particular concentration limits, by the U.S. Food and Drug Administration (FDA).

Phenolic compounds containing polyhydroxyl moieties are usually accountable for overall antioxidant activity. These components have great capability to scavenge and prevent lipid peroxidation. In the present research, we incorporated GEO in CS-POR composite polymeric material to improve the shelf life and prevent oxidation of food products. Results obtained from total phenolic assay revealed that GEO had (5.97 ± 0.01) mg of gallic acid equivalents per g of dry mass of the sample of total phenols. As demonstrated in Figure 7 and Figure 8, S1 films containing only chitosan (90% DA) showed minimal antioxidant effects, contradicting the previous reports suggesting the antioxidant effects of CS [5,20]. However, this could be due to the change in degree of deacetylation, molecular weight, purity of CS, as well as type of the method used for determining antioxidant activity. The S2 film, which was plasticized using glycerol, showed comparable antioxidant effects to S1, which contradicts the previous study where glycerol at 2.5% showed antioxidant effects against the hydroxyl radical however failed to show antioxidant effects against chlorine monoxide or peroxynitrite radicals [21]. This could be due to the strong molecular interaction between CS and glycerol, which decreased the availability of its functional groups to show antioxidant effects. Figure 7 and Figure 8 showed that S5 film demonstrated almost equivalent antioxidant effects, e.g., standards (Trolox and butylated hydroxyanisole). Overall findings obtained from antioxidant assays showed the significant antioxidant activity of S5, which could be due to the synergistic interactions between phenolic components of GEO and POR. On other hand, CS (S1) and CS (S2) plasticized films showed less antioxidant effects than (S3–S5). GEO loaded S3 and POR containing S4 samples showed almost comparable antioxidant effects, suggesting that both GEO and POR have antioxidant potential.

## 3. Materials and Methods

### 3.1. Chemicals

Ginger oil with specifications including pale yellow to brown color, mobile liquid appearance, aromatic odor, 1.481–1.499 refractive Index, 0.896–0.912 specific gravity −28 flash point 99 °C optical rotation, Batch no. NNIGIEO/104/0821, 100 mL was purchased from Nature Natural, India, Chitosan (extrapure, 150–500 m·Pas, 90% DA by SRL Pvt. Ltd., Mumbai, India), glycerol (BDH Laboratory, London, England), and tween 80 (Merck KGaA, Darmstadt, Germany). Acetic acid, ferric chloride, potassium persulfate, 2,2′-diphenyl-1-picrylhydrazyl (DPPH), trolox (6-Hydroxy-2,5,7,8-tetramethylchromane-2-carboxylic acid), butylated hydroxyl anisole, ABTS (2,2′-azinobis-(3-ethylbenzothiazoline-6-sulfonic acid)) were purchased from Sigma-Aldrich (St. Louis, MO, USA).

### 3.2. Isolation of Porphyran from Pyropia vietnamensis

The polysaccharide fraction from *Pyropia vietnamensis* was isolated by using a procedure used in our previous work [2].

### 3.3. Edible Films (EFs) Production

Casting procedure was used to fabricate CS and POR based coating material. Five different types of films (S1–S5) were developed, two of which (S3 and S5) were loaded with GEO and Gly while the first one (S1) was used as a CS control without GEO and plasticizer. Initially, 1% (*w*/*v*) of CS (90% DA) was added to acetic acid (1%, *v*/*v*) solution. The resultant solution was stirred overnight at room temperature via magnetic stirrer until completely dissolved. Once CS was completely solubilized, the resultant solution was divided equally into five different beakers (50 mL) and labeled as S1–S5. CS film-forming solution without glycerol and GEO in the first beaker coded as S1 was used as the blank group. To the CS solution in second beaker coded as S2, glycerol (0.5%) was added whereas in third beaker coded as S3, glycerol (0.5%, *v*/*v*), Tween 80 (1.0%, *w*/*v*), and GEO (1%, *v*/*v*) were added to prepare 1% CS GEO, film-forming solution. In the fourth beaker coded as S4, glycerol (0.5%, *v*/*v*) and 1% (*w*/*v*) of POR were added gradually and stirred overnight at room temperature via magnetic stirrer. In the last beaker coded as S5, glycerol (0.5%, *v*/*v*), Tween 80 (1.0%, *w*/*v*), and 1% (*w*/*v*) of POR and GEO (1%, *v*/*v*) were added. Film-forming solution present in the beakers (S1–S5) was transferred into plastic petri plates and kept for drying at room temperature for 48 h. After drying, all films were observed and peeled off from the plastic petri plate surface (Figure 2). At this stage, all these films were characterized based on their visual appearance. The prepared films were kept in plastic bags and held in desiccators at 50% relative humidity for 24 h before further experiments. All treatments were performed in triplicate.

### 3.4. Thickness of the EFs

The thickness of the fabricated EFs was calculated using a handheld micrometer (Mitutoyo digital micrometer 2046F, Mitutoyo, Kawasaki, Japan) with a precision of ±1 µm in 10 replicas for each film randomly. Mean values were determined in mm.

### 3.5. Mechanical Testing

The mechanical strength of the films was assessed by measuring the elongation at break (EB) as well as tensile strength (TS) of the fabricated EFs using the TA. A HD Plus Texture Analyzer (Stable Micro Systems Ltd., Godalming, UK) was used as per ASTM 93 D882-2010 standard [22]. After preconditioning at 70% relative humidity up to 24 h, film samples were cut into rectangular strips of 20 mm × 80 mm. Films were analyzed at the strain rate of 40 mm/min. The test was executed in a triplicate manner and then the average value was determined. The tensile strength (TS) in MPa was calculated as follows:*TS* = *Mf*/*CS*
where *Mf* signifies maximum force at break (N) and *CS* was the area of EF cross-section (m × m).

Elongation at break (*EB*) was calculated by using equation:*EB* = [(*L*_1_ − *L*_2_)/*L*_2_] *×* 100
where *EB* signifies elongation at break (%), *L*_2_ represents the dimension of the EF when it breaks (mm), and *L*_1_ was the previous dimension of the EF.

### 3.6. Water Vapor Permeability (WVP)

The WVP of the films was determined gravimetrically at 25 °C in accordance with the work of Zhou et al. with some changes [1]. Selected films without any perforation or defects were cut into circular shape with a size somewhat greater than the size of the mouth of a small Erlenmeyer flask made of glass. Before covering with film disc, the glass flask was fully loaded with anhydrous calcium chloride (5 g, 0% RH) and then covered with a disc followed by sealing it with paraffin. Sealed flasks were placed in a desiccator containing NaCl solution (75% relative humidity) at 25 °C. Calcium chloride was filled up to a level at which an air space of 1 cm between the film underside and the desiccant must be left. The relative humidity in the flask was lesser than outside. WVP was calculated from the weight gain due to the transport of water vapors from outside to inside the environment at a steady state. The change in the weight of the flasks (weight loss) over time (6 times at two hours interval prior to a steady state was attained) was determined, where the slope was evaluated by linear regression using Microsoft^®^ Office Excel 2021 (R^2^ 0.998). Testing was performed in a triplicate manner with each sample. WVP was determined as per the following equation:*P = (*Δ*W* × *S)/(A*_2_ × *T* × Δ*WP*)
where *P* = WVP (×10^−12^ g cm cm^2^ s^−1^ Pa^−1^); Δ*W* = difference in weight of flask over a period (determined from the slope of the curve); *S* = mean thickness of the film (cm); *A*_2_ = effective area of the films (cm^2^); *T* = mean time (in seconds), and Δ*WP* = water vapor partial pressure difference (atm) between over both sides of the films.

### 3.7. Oxygen Barrier Property

Oxygen permeability (OP) of the prepared EFs, which is related to camellia oil peroxidation, was measured via titration with standard sodium thiosulphate using the method proposed by Kurt et al. [23] with minor changes. As per this procedure, 1.5 mL of camellia oil was poured into a 5 mL glass test tube and covered with EF followed by sealing with paraffin. Prepared samples were stored for 7 days at room temperature with RH of 50%. Oxidation of camellia oil was determined by using sodium thiosulfate titration (Na_2_S_2_O_3_). Oxygen permeability was determined by using following equation:*OP* (*in g*/100 *g*) = [(*F* − *B*) × *S* × 0.1269]/*W* × 100
where *F* stands for amount of standard sodium thiosulphate solution (mL) consumed by the film and *B* stands for volume of standard sodium thiosulphate solution (mL) consumed by the blank. *S* means the concentration of standard sodium thiosulphate (mol/L), 0.1269 signifies the value of iodine equal to one milliliter of standard Na_2_S_2_O_3_ solution (g), and *W* is the value of the EF (g) and 100 signifies conversion factor.

### 3.8. Water Solubility and Swelling Degree

The procedure proposed by Souza et al. (2017) was used with slight modification to determine the water solubility and swelling degree [24]. Films (S1–S5) were cut into 10 × 40 mm pieces and were weighed in the analytical balance to obtain initial weight (*W*_1_) of the samples, followed by drying in a desiccator to obtain initial dry mass (*W*_2_). Later, the dried sample was placed in 25 mL water (Milli-Q), and kept at 25 ± 2 °C for 24 h to facilitate the swelling process. Afterwards, EFs were subjected to drying by using filter paper and weighted (*W*_3_) again. Final samples were dried to a constant weight at 60 °C in an oven to measure the final dry mass of the films. Water solubility and swelling degree were calculated as follows:*S* = (*W*_2_ − *W*_4_)/*W_i_* × 100
*S* = (*W*_3_ − *W*_2_)/*W_i_* × 100
where *S* stands for water solubility (%)

### 3.9. Moisture Content

To determine the moisture content of prepared films (S1–S5) of size 2 × 2 cm, films were initially weighed and placed in an empty glass petri dish. Later, glass petri dishes containing EFs were placed for drying in a hot air oven (without vacuum) for 24 h at 90 °C till a constant dry mass was achieved and weight loss was determined as per the equation mentioned below:*MC* = (*M*_1_ − *M*_2_)/*M*_2_
where *MC* was moisture content, *M*_1_ was the initial film weight (mg), and *M*_2_ was the final weight (mg).

### 3.10. Optical Transmittance Analysis

Film resistance to UV and visible light (200–800 nm) represents the transmittance potential of the EFs which was assessed by UV–V is spectrophotometer (UV160U-Shimadzu, Japan). The test EFs were cut into shape with dimensions of 5 × 20 mm and placed to the inner side of a cuvette and a cuvette without film (blank) was considered for reference. The transparency of each test sample was measured three times and calculated as per the method proposed by Shiku et al. [25].

### 3.11. Color Measurement of the EFs

The color attributes (L*, a*, b*, and ΔE*) of the EFs were measured by chromameter (CR-400, Konica Minolta, Tokyo, Japan) and, prior to taking sample readings, the instrument was standardized using white and black tiles. This is measured on the basis of the CIE L*, a*, b* color system, where L indicates lightness, a indicates red-green color, and b shows yellow-blue color. For measurement, 5 different points were selected on the surface of EFs and average value was obtained. The overall color variation (ΔE*) between the color parameter of the respective film samples and that of white standard plate (*L** = 74.11, *a** = 1.11, and *b** = 0.47) were analyzed by using equation as mentioned below:*ΔE** = [(Δ*L**)^2^ + (Δ*a**)^2^ + (Δ*b**)^2^]^1/2^
and chroma intensity (*CI*) was determined by using following equation:*CI* = (*a*^2^* + *b*^2^*)^1/2^

### 3.12. Thermogravimetric (TGA) Analysis

The thermal stability of EFs was determined using a thermal gravimetric (TG) analyzer (TA instruments, New Castle, DE, USA, SDT-Q600) in a nitrogen atmosphere. EFs were scanned in the temperature range of 25–600 °C at a heating rate of 10 °C/min.

### 3.13. X-ray Diffraction (XRD) Studies

XRD patterns of the EFs were characterized using a Bruker D8 Discover instrument operated at 40 kV with 2θ ranging from 5 to 50°.

### 3.14. SEM (Scanning Electron Microscopy) Analysis

Surface as well as cross-sectional morphologies of the S1–S5 films were investigated using SEM at 20 kV using a JSM6510LA, Analytical SEM, Jeol, Japan. The S1–S5 films were fractured manually in liquid nitrogen, then mounted on aluminum stubs with adhesive tape and sputter-coated by a thin layer of gold prior to taking images.

### 3.15. FTIR Spectra Analysis

The chemical structures of EFs were characterized by using FTIR Spectrometer (InfraRed Bruker Tensor 37, Ettlingen, Germany) set up with an attenuated total reflection (horizontal) device (45o ZnSe) over a range of 400 cm^−1^ to 4000 cm^−1^ with a resolution of 4 cm^−1^ and 32 scans.

### 3.16. Antioxidant Assays

#### 3.16.1. Sample Preparation

Samples (500 mg) in the form of edible films (EFs) were weighed with the addition of methanol (15 mL). EFs solutions were initially mixed vigorously by using a vortex mixer (VWR int., Germany) followed by centrifugation at 10,000 RPM for 5 min at 5 °C. The supernatant obtained was utilized for assessing antioxidant activities.

#### 3.16.2. Assessment of Total Phenolic Content

The assessment of phenolic level was performed as per the earlier reported method. The supernatant solutions (S1–S5) obtained were mixed with desired concentration of Folin and Ciocalteu’s reagent. The resultant solution was incubated at room temperature with continuous stirring. An equal proportion of sodium bicarbonate was added, and absorbance was noted utilizing microplate reader using blank sample at 517 nm. The results obtained were expressed as mg/gm of GAE (gallic acid equivalent). All the tests were repeated thrice.

#### 3.16.3. Determination of Scavenging Capacity against DPPH Radical

The scavenging activity of samples against DPPH radical was determined as per earlier method [26]. The supernatant solutions (S1–S5) obtained were mixed with an equal quantity of DPPH solution in 96-well microplates. The reaction mixture was incubated for 30 min at 37 °C with continuous stirring. The absorbance was measured using a microplate reader using a blank sample at 517 nm. The butylated hydroxy anisole was used as a standard. All the tests were repeated thrice. The percentage inhibition was calculated as:*Percentage inhibition* (%) = (*Ac* − *At*)/(*Ac*) × 100
where *Ac* = absorbance of the control solution and *Atc* = absorbance of test solution.

#### 3.16.4. Analysis of Trolox Equivalent Antioxidant Capacity (TEAC)

Analysis of TEAC was performed as per an earlier reported method [27]. In this method, the quantity of ABTS (7 mM) was mixed with 2.45 mM of K_2_S_2_O_8_ (potassium persulfate). The resultant mixture was incubated at room temperature for 12 h. The ABTS^•+^ radical generated was then mixed with alcohol and absorbance was measured at 734 nm. Then, the supernatant solutions (S1–S5) obtained were mixed with ABTS^•+^. The mixture was again incubated, and absorbance was measured as above. The results obtained were expressed as μM Trolox equivalents per gram of film. All the tests were repeated thrice.

#### 3.16.5. Determination of FRAP (Ferric Antioxidant Reducing Power)

FRAP was performed as per a method reported earlier [28]. In this method, supernatant solutions (S1–S5) obtained were mixed with an equal quantity of FRAP solution. The resultant mixture was then incubated, and absorbance was recorded. Final readings obtained were expressed as μM Trolox equivalents per gram of film. All the experiments were performed in a triplicate manner.

#### 3.16.6. Statistical Analysis

All results are expressed as the mean value ± standard deviation (S.D.) of three independent replicates. One-way analysis of variance followed by Duncan’s test was performed to test the significance of differences between mean values at the 5% level of significance using statistical analysis software.

## 4. Conclusions

CS based EFs have considerable demerits, such as poor mechanical and barrier properties along with greater sensitivity against moisture, which restricts their utilization as food coating materials. The development of composite films loaded with safe and natural antioxidants could be an interesting approach to overcome these challenges offered by blank CS. FTIR spectral assessment revealed that GEO was successfully integrated into the CS-POR film. Overall findings suggested that composite CS-POR-GEO EFs (S5) showed an increase in thickness, TS, and EB with a decrease in YM, WVP, OP, WS, SD, MC, and transparency. TGA and XRD analyses of S4 and S5 composite films showed greater thermal stability and less crystallinity than S1–S3 films. SEM analysis and visual inspection revealed that S4 and S5 films showed a more homogenous, uniform appearance, with no cracks, and pores, and require minimum effort to peel compared to S1–S3. Additionally, the antioxidant effects of blended films were enhanced due to the incorporation of POR and GEO. The current investigation showed the potential of POR and GEO in overcoming the demerits of CS films. It was found that the GEO loaded CS-POR composite showed better physical-chemical properties with the antioxidant potential almost equivalent to the standards (Trolox and butylated hydroxyanisole) used in the study. Thus, CS-POR-GEO EFs could be considered as a potential alternative food packaging material to CS EFs.

## Figures and Tables

**Figure 1 polymers-14-01782-f001:**
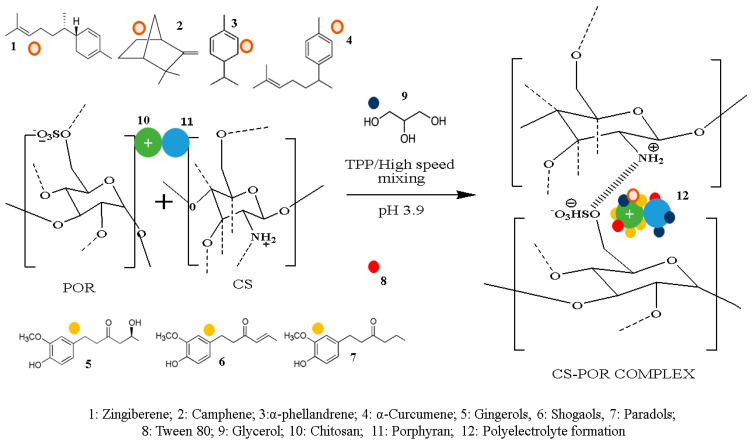
Polyelectrolyte complex (PEC) formation between chitosan (CS) and porphyran (POR) under the specified experimental conditions (pH 3.9).

**Figure 2 polymers-14-01782-f002:**
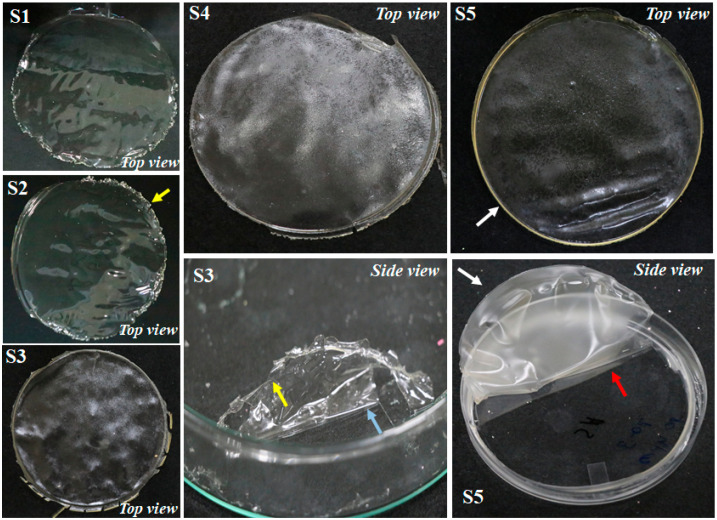
Visual appearances of CS blank film (S1), CS-Gly film (S2), CS-S-GEO film (S3), CS-POR-Gly film (S4), and CS-POR-Gly-GEO film (S5); yellow arrows indicate film crumble into pieces during peeling process; blue arrow represents more force was required for peeling; white arrows represent films were not broken into pieces during peeling; red arrow represents less force required for peeling.

**Figure 3 polymers-14-01782-f003:**
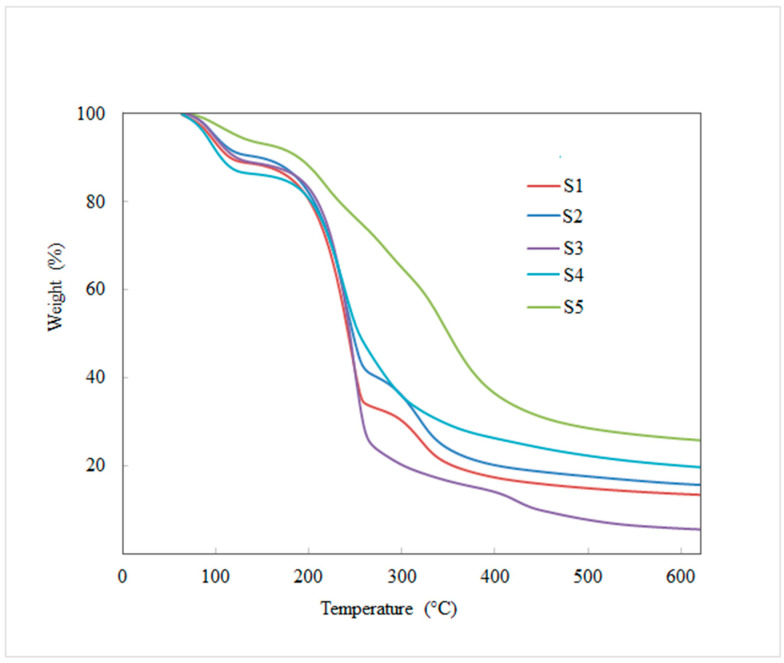
Thermogravimetric curves of CS blank film (S1), CS-Gly film (S2), CS-Gly-GEO film (S3), CS-POR-Gly film (S4), and CS-POR-Gly-GEO film (S5).

**Figure 4 polymers-14-01782-f004:**
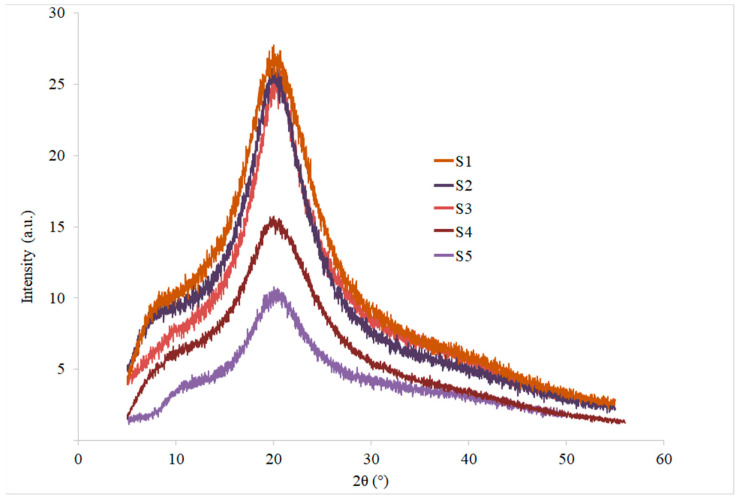
XRD patterns of CS blank film (S1), CS-Gly film (S2), CS-Gly-GEO film (S3), CS-POR-Gly (S4), and CS-POR-Gly-GEO film (S5).

**Figure 5 polymers-14-01782-f005:**
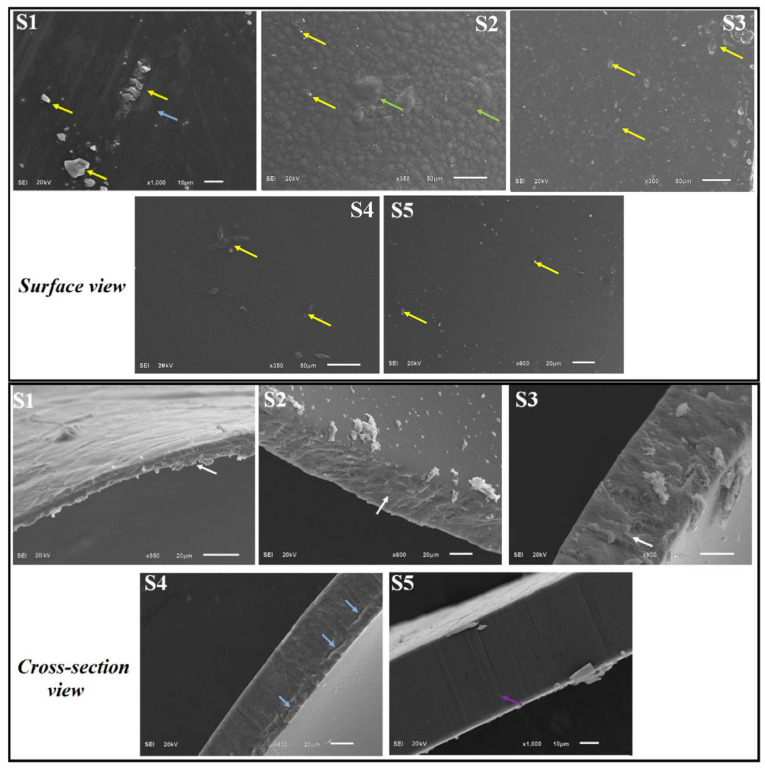
SEM images of CS blank film (S1), CS-Gly film (S2), CS-Gly-GEO film (S3), CS-POR-Gly film (S4), and CS-POR-Gly-GEO film (S5) (superior and cross section view); yellow arrows represent particles over the surface; purple colour arrows indicate compact, uniform, homogenous, and dense structure; green colour represents bulges over the surface and roughness; white arrows represent porosity and irregularity in the film; and blue arrows represents the cracks over the surface and cross-section.

**Figure 6 polymers-14-01782-f006:**
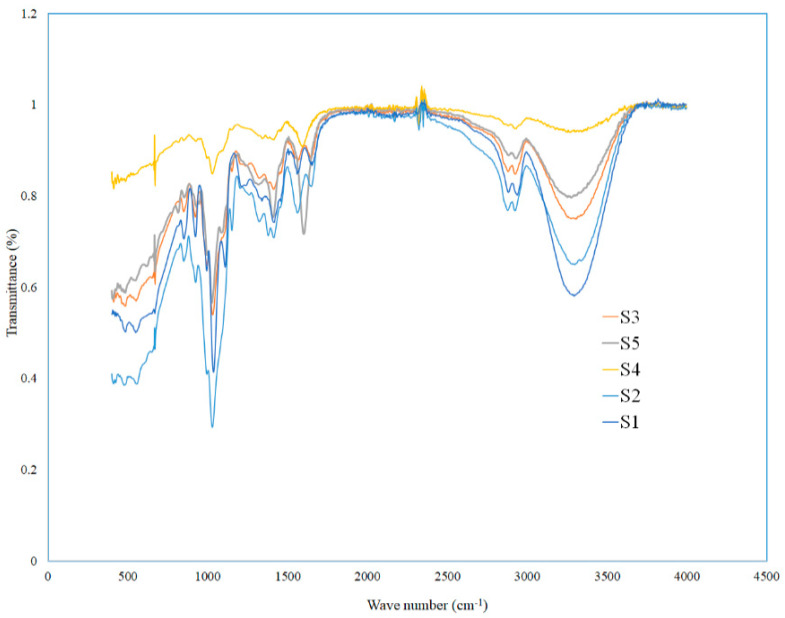
FTIR spectra of CS blank film (S1), CS-Gly film (S2), CS-Gly-GEO film (S3), CS-POR-Gly film (S4), and CS-POR-Gly-GEO film (S5).

**Figure 7 polymers-14-01782-f007:**
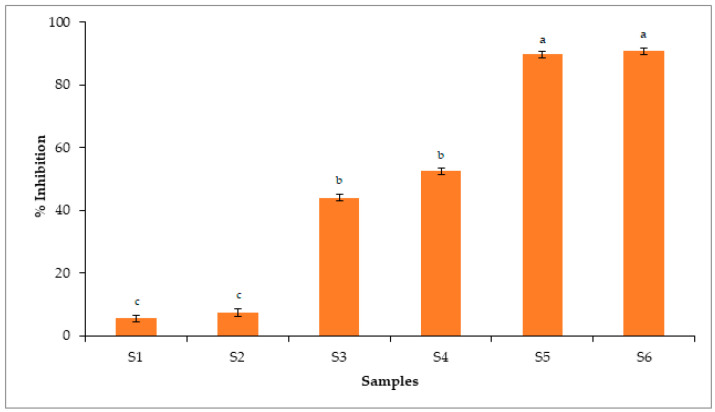
Antioxidant capacity of CS blank film (S1), CS-Gly film (S2), CS-S-GEO film (S3), CS-POR-Gly film (S4), CS-POR-Gly-GEO film (S5), butylated hydroxyl anisole (S6) determined by DPPH assays. Bars represent mean values ± standard deviations. Bar with different letters (a, b & c) shows significant difference among the groups (*p* < 0.05).

**Figure 8 polymers-14-01782-f008:**
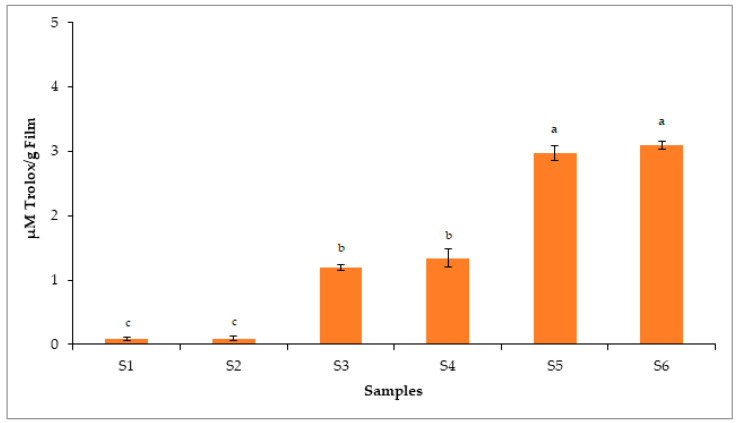
Antioxidant potential of CS blank film (S1), CS-Gly film (S2), CS-S-GEO film (S3), CS-POR-Gly film (S4), CS-POR-Gly-GEO film (S5) and Trolox (S6) determined by TEAC assay. Bars represent mean values ± standard deviations. Bar with different letters (a, b & c) shows significant difference among the groups (*p* < 0.05).

**Table 1 polymers-14-01782-t001:** General characterization of EFs.

EF Codes	Composition	Appearance of Films
S1	CS	Transparent, fragile, brittle, stiff, inflexible, difficult to peel, non-sticky
S2	CS-Gly	Less transparent than S1, brittle and fragile, flexible than S1, easy to peel and handle
S3	CS-S-EO	Less transparent than S1 and S2, sticky, brittle and fragile, more flexible than S1 and S3, difficult to peel and handle,
S4	CS-POR-Gly	Less transparent than S1 and S2, not brittle and fragile, not rigid, flexible, easy to peel, less sticky
S5	CS-POR-Gly-EO	Less transparent than S1–S4, not fragile and brittle, easy to peel off from the surface, more flexible than S1–S4, not adhesive

**Table 2 polymers-14-01782-t002:** WVP, thickness, mechanical properties and OP (oxygen permeability) of the EFs.

Formulations	WVP (×10^−12^ g⋅cm/cm^2^⋅s⋅Pa)	Thickness (μm)	EB (%)	TS (MPa)	Young Modulus (MPa)	OP (g/100 g)
S1	2.3 ± 0.06 ^a^	52.12 ± 2.3 ^a^	2.17 ± 1.71 ^a^	21.23 ± 7.11 ^a^	77.21 ± 0.51 ^a^	4.17 ± 0.034 ^a^
S2	2.9 ± 0.02 ^b^	55.67 ± 1.7 ^b^	17.12 ± 4.21 ^b^	16.24 ± 2.71 ^b^	52.12 ± 0.54 ^c^	3.21 ± 0.030 ^b^
S3	2.1 ± 0.03 ^a^	58.12 ± 2.1 ^c^	37.24 ± 9.31 ^c^	11.24 ± 3.16 ^c^	42.16 ± 0.71 ^b^	2.11 ± 0.037 ^c^
S4	1.6 ± 0.01 ^c^	46.21 ± 3.7 ^d^	33.27 ± 6.81 ^c^	41.32 ± 1.24 ^d^	38.42 ± 0.42 ^b^	1.78 ± 0.026 ^c^
S5	1.1 ± 0.05 ^d^	49.78 ± 1.8 ^e^	71.12 ± 12.3 ^d^	32.77 ± 6.71 ^e^	27.21 ± 0.21 ^d^	0.77 ± 0.011 ^d^

Results are expressed as mean value ± standard deviation. Values with numerous superscript symbols in each row suggest substantial variations between the samples (*p* < 0.05).

**Table 3 polymers-14-01782-t003:** WS and SD of the EFs.

Codes of the Samples	WS (%)	Swelling Degree (%)	MC (%)
S1	26.1 ± 0.7 ^a^	167 ± 12.2 ^a^	12.71 ± 0.12 ^a^
S2	34.2 ± 1.1 ^b^	198 ± 17.7 ^b^	18.46 ± 0.11 ^b^
S3	23.2 ± 0.6 ^c^	112 ± 9.1 ^c^	15.62 ± 0.17 ^c^
S4	21.4 ± 3.2 ^c^	79 ± 4.2 ^d^	15.27 ± 0.11 ^c^
S5	12.2 ± 0.9 ^d^	37 ± 7.1 ^e^	14.81 ± 0.12 ^c^

Results are expressed as mean value ± standard deviation. Values with numerous superscript symbols in each row suggest substantial variations between the samples (*p* < 0.05).

**Table 4 polymers-14-01782-t004:** Color measurements and transparency of the EFs.

	L	a*	b*	△E*	CI*	Transparency (%)
S1	67.22 ± 3.01 ^a^	1.17 ± 0.03 ^a^	0.31 ± 0.02 ^a^	6.81 ± 3.17 ^a^	1.20 ± 0.21 ^a^	42.81 ± 1.24 ^a^
S2	62.43 ± 2.84 ^b^	1.64 ± 0.01 ^b^	0.77 ± 0.01 ^b^	12.6 ± 1.34 ^b^	1.81 ± 0.13 ^b^	33.23 ± 2.24 ^b^
S3	41.18 ± 5.22 ^c^	−5.24 ± 0.32 ^c^	4.23 ± 0.22 ^c^	33.4 ± 7.31 ^c^	6.72 ± 0.34 ^c^	30.22 ± 1.67 ^c^
S4	63.56 ± 4.32 ^b^	1.62 ± 0.07 ^b^	0.71 ± 0 07 ^b^	10.5 ± 2.88 ^b^	1.76 ± 0.52 ^b^	26.75 ± 1.07 ^d^
S5	38.30 ± 6.21 ^d^	−6.78 ± 0.55 ^d^	4.77 ± 1.13 ^c^	36.5 ± 6.24 ^d^	8.21 ± 0.64 ^d^	28.34 ± 3.21 ^d^

Results are expressed as mean value ± standard deviation. Values with numerous superscript symbols in each row suggest substantial variations between the samples (*p* < 0.05); L: lightness, a: red-green color, b: yellow-blue color; ΔE*: overall color variation; CI: chroma intensity.

## Data Availability

The data will be available from the corresponding author following reasonable request.

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
