# Peer review of "Development and Characterization of Chitosan and Porphyran Based Composite Edible Films Containing Ginger Essential Oil"

_polymers, 2022, doi:10.3390/polym14091782_

Round 1

Reviewer 1 Report

The manuscript describes an interesting strategy to prepare an alternative food packaging material and the study was well designed. In addition, the data is quite adequate and interesting due to potential practical application. However, the language needs to be extensively improved as it is not that easy to follow.

Minor Comments

  1. In the abstract section, please revise the following sentences;

……..assay suggested the present of high………...  

…………thus can be used as potential a packaging material in food industry.

  1. In Figure 3 (TGA), sample 5 showed significant deviation as compared to other formulations tested, while in Figure 4 (XRD), the thermogram did not show any clear effect.
  2. The figure legends are also not very clear in figure 4 to differentiate the thermograms.
  3. Figures 7 and 8; describe a, b, and c mentioned in the bar.
  4. Please thoroughly revise the English by a native English speaker.

Author Response

Dear Reviewer,

Thank you for giving me the opportunity to submit a revised draft of my manuscript titled “Development and Characterization of Chitosan and Porphyran Based Composite Edible Films Containing Ginger Essential Oil” to Polymers, MDPI. I appreciate the time and effort that you and the reviewers have dedicated to providing your valuable feedback on my manuscript. I am grateful to the reviewers for their insightful comments on my paper. I have been able to incorporate changes to reflect most of the suggestions provided by the reviewers. I have highlighted the changes within the manuscript and submit it accordingly.  Here is a point-by-point response to the reviewers’ comments and concerns.

The manuscript describes an interesting strategy to prepare an alternative food packaging material and the study was well designed. In addition, the data is quite adequate and interesting due to potential practical application. However, the language needs to be extensively improved as it is not that easy to follow.

Comment 1: In the abstract section, please revise the following sentences.

. ……..assay suggested the present of high………...

…………thus can be used as potential a packaging material in food industry.

Response: Thank you for this valuable feedback. These are revised now, and all corrections are highlighted in the manuscript.

Comment 2: In Figure 3 (TGA), sample 5 showed significant deviation as compared to other formulations tested, while in Figure 4 (XRD), the thermogram did not show any clear effect.

Response: XRD patterns at room temperature represents the extent of crystallinity and amorphousness of the films whereas TGA represents the thermal decomposition as well as stability of the films. XRD patterns represented in the Figure 4 showed the extent of crystallinity and amorphousness.  As shown in the figure 4 and mentioned in text that the crystallinity of the CS-POR based films (S4, S5) was significantly lower than that of the CS based films (S1-S3). Moreover, XRD overlay represents the effect of GEO over the crystallinity of the films. Incorporation of GEO reduced the crystallinity of the films. It was also suggested that the crystallinity of the CS-POR based films (S4, S5) was significantly lower than that of the CS based films (S1-S3). Furthermore, crystallinity of the EFs with GEO was considerably less than that of the EFs without GEO under the similar experimentation conditions. This suggested that incorporation of GEO may decrease the CS-POR crystallinity. The decrease in intensity of the peak could be associated with intermolecular interaction between CS-POR, resulting in reduction in the mobility of molecules and thus preventing crystallization. This justification is already included in the manuscript.

Comment 3: The figure legends are also not very clear in figure 4 to differentiate the thermograms.

Response: Thanks for pointing out this. Presentation of Figure legends in the figure 4 is now improved. Now the legends are modified (legends lines width is now increased to improve their visibility).

Comment 4: Figures 7 and 8; describe a, b, and c mentioned in the bar.

Response: Thanks for your valuable feedback. Bar with different letters (a, b & c) shows significant difference among the groups (p<0.05). This information is included and highlighted in the text.

Comment 5: Please thoroughly revise the English by a native English speaker.

Response: We regret there were problems with the English. Now the paper has been carefully revised by a native English speaker to improve the grammar and readability.

In addition to the above comments, all spelling and grammatical errors pointed out by the reviewers have been corrected. We look forward to hearing from you in due time regarding our submission and to respond to any further questions and comments you may have.

Sincerely

Reviewer 2 Report

In this manuscript, the authors assess the optical, mechanical, antioxidative and thermal properties of composite materials made from polyelectrolyte complex of chitosan with porphyran, with ginger essential oil as an additive. The authors perform extensive characterization of the material, showing trade-offs in color deterioration vs. improved antioxidative properties and oxygen permeability, moduli vs. work of fracture, etc. I recommend publication of this manuscript once the following minor points are addressed:

  • Some of the abbreviations are only described in the Methods section after it already appeared in the Discussion section, and are inconsistent. For example, it would increase readability if the author introduced what EB, Y and TS are when they first appear. Additionally, at line 514 the elongation at break is abbreviated as EAB, possibly by mistake.
  • Equations such as lines 520, 524 and 543 should be properly formatted with italicized and properly scripted variables.
  • It is not clear to me what the authors meant in line 330-331: "Out of all prepared EFs, S3 and S5 films containing GEO showed thermal decomposition at initial temperature at 57C." I see that all measurements start at that point as 100% in Figure 3: do they mean that the mass loss is already happening before 57C? Please clarify.
  • The SEM analysis portion of this paper is somewhat rudimentary. It is not clear how the samples were prepared: given the differences in mechanical properties, creating the fracture surface at room temperature for these samples can lead to different textures and degree of artifact introduction. It may be helpful to create the fracture surface by fracturing in liquid nitrogen. Also, the surface particles (yellow) can come from any contaminants in the sample preparation steps. I would limit discussion on what can be seen as obvious differences without sample prep artifacts, possibly on surface roughness in S2 and cracks seen in S3. 

Author Response

Dear Reviewer,

Thank you for giving me the opportunity to submit a revised draft of my manuscript titled “Development and Characterization of Chitosan and Porphyran Based Composite Edible Films Containing Ginger Essential Oil” to Polymers, MDPI. I appreciate the time and effort that you and the reviewers have dedicated to providing your valuable feedback on my manuscript. I am grateful to the reviewers for their insightful comments on my paper. I have been able to incorporate changes to reflect most of the suggestions provided by the reviewers I have highlighted the changes within the manuscript and submit it accordingly. Here is a point-by-point response to the reviewers’ comments and concerns.

In this manuscript, the authors assess the optical, mechanical, antioxidative and thermal properties of composite materials made from polyelectrolyte complex of chitosan with porphyran, with ginger essential oil as an additive. The authors perform extensive characterization of the material, showing trade-offs in color deterioration vs. improved antioxidative properties and oxygen permeability, moduli vs. work of fracture, etc. I recommend publication of this manuscript once the following minor points are addressed:

Comment 1: Some of the abbreviations are only described in the Methods section after it already appeared in the Discussion section and are inconsistent. For example, it would increase readability if the author introduced what EB, Y and TS are when they first appear. Additionally, at line 514 the elongation at break is abbreviated as EAB, possibly by mistake.

Response: Thanks for your valuable feedback and sorry for this mistake. Initially manuscript was prepared, considering material and methods comes first followed by the results and discussion.  That’s why most of the abbreviations are included in material and methods. Now its corrected and highlighted.

Comment 2: Equations such as lines 520, 524 and 543 should be properly formatted with italicized and properly scripted variables.

Response: Thanks for your valuable suggestion. Equations are now italicized and properly scripted.

Comment 3: It is not clear to me what the authors meant in line 330-331: "Out of all prepared EFs, S3 and S5 films containing GEO showed thermal decomposition at initial temperature at 57C." I see that all measurements start at that point as 100% in Figure 3: do they mean that the mass loss is already happening before 57C? Please clarify.

Response: Thanks for your valuable suggestion. Yes, first weight loss was observed for all films at 57C. Its corrected and highlighted now as follows:  All prepared EFs, showed first weight loss at temperature of 57◦C.

Comment 4: The SEM analysis portion of this paper is somewhat rudimentary. It is not clear how the samples were prepared: given the differences in mechanical properties, creating the fracture surface at room temperature for these samples can lead to different textures and degree of artifact introduction. It may be helpful to create the fracture surface by fracturing in liquid nitrogen. Also, the surface particles (yellow) can come from any contaminants in the sample preparation steps. I would limit discussion on what can be seen as obvious differences without sample prep artifacts, possibly on surface roughness in S2 and cracks seen in S3. 

Response: Thanks for your suggestion. The films were fractured manually in liquid nitrogen and then mounted on aluminium stubs with adhesive tapes and sputter-coated by a thin layer of gold prior to taking images. This procedure was used and included now in the manuscript.

In addition to the above comments, all spelling and grammatical errors pointed out by the reviewers have been corrected. We look forward to hearing from you in due time regarding our submission and to respond to any further questions and comments you may have.

Sincerely

Reviewer 3 Report

Lines  68-69. "...CS and POR forms a stable stoichiometric complex via forming a strong electro- 68 statically bond between amine group of CS and sulphate group of POR..." Perhaps the authors should use the term "salt" instead of "complex"

Line 81. The authors mention that ginger oil contains phenolic substances, which are responsible for the necessary properties of the resulting material, but Figure 1 shows quite different substances and no phenolic substances. There is an additional question to this same figure, which is not mentioned by the authors in the article. The point is that the terpenoids depicted are volatile and have a specific odor, which can affect the product packaged in the material. Indeed, if we are talking about the packaging of gingerbread, then the presence of volatile ginger fragrances in the packaging will only benefit them, but if we are talking about products that could be harmed by such flavoring, it is at least worth discussing.

Lines 478-479, 485-486. The description of the materials used does not allow us to reproduce the study described, nor does it allow us to get an idea of what materials were used. Since ginger oil, chitosan, and porphyran are materials of natural origin, they are characterized by a high variability in composition and structure, molecular weights and their distribution, etc. For ginger oil, it would be useful to give at least the data of chromato-mass spectrometric analysis; for polysaccharides, a description of the original source of the materials and gel-exclusion chromatograms are necessary so that the reader can imagine exactly what materials were used. Indeed, for the practical use of polysaccharides, one of the main obstacles is the poor reproducibility in their preparation, insufficiently adequate analysis of the synthesized material, which often makes it impossible to reproduce the result obtained.

Author Response

Dear Reviewer,

Thank you for giving me the opportunity to submit a revised draft of my manuscript titled “Development and Characterization of Chitosan and Porphyran Based Composite Edible Films Containing Ginger Essential Oil” to Polymers, MDPI. I appreciate the time and effort that you and the reviewers have dedicated to providing your valuable feedback on my manuscript. I am grateful to the reviewers for their insightful comments on my paper. I have been able to incorporate changes to reflect most of the suggestions provided by the reviewers. I have highlighted the changes within the manuscript and submit it accordingly.  Here is a point-by-point response to the reviewers’ comments and concerns.

Comment 1: Lines 68-69. "...CS and POR forms a stable stoichiometric complex via forming a strong electro- 68 statically bond between amine group of CS and sulphate group of POR..." Perhaps the authors should use the term "salt" instead of "complex"

Response: Thanks for your valuable suggestion. Perhaps its complex as mentioned in following pervious reports. Still, we are ready to change it if reviewer will disagree with this. 

  • Jamróz E, Janik M, Juszczak L, Kruk T, Kulawik P, Szuwarzyński M, Kawecka A, Khachatryan K. Composite biopolymer films based on a polyelectrolyte complex of furcellaran and chitosan. Carbohydr Polym. 2021 Nov 15;274:118627. doi: 10.1016/j.carbpol.2021.118627. Epub 2021 Sep 1. PMID: 34702453.
  • Kulig D, Zimoch-Korzycka A, Jarmoluk A, Marycz K. Study on Alginate⁻Chitosan Complex Formed with Different Polymers Ratio. Polymers (Basel). 2016 May 4;8(5):167. doi: 10.3390/polym8050167. PMID: 30979272; PMCID: PMC6432350.
  • Dubashynskaya NV, Raik SV, Dubrovskii YA, Demyanova EV, Shcherbakova ES, Poshina DN, Shasherina AY, Anufrikov YA, Skorik YA. Hyaluronan/Diethylaminoethyl Chitosan Polyelectrolyte Complexes as Carriers for Improved Colistin Delivery. Int J Mol Sci. 2021 Aug 4;22(16):8381. doi: 10.3390/ijms22168381. PMID: 34445088; PMCID: PMC8395075.
  • Potaś J, Szymańska E, Wróblewska M, Kurowska I, Maciejczyk M, Basa A, Wolska E, Wilczewska AZ, Winnicka K. Multilayer Films Based on Chitosan/Pectin Polyelectrolyte Complexes as Novel Platforms for Buccal Administration of Clotrimazole. Pharmaceutics. 2021 Sep 30;13(10):1588. doi: 10.3390/pharmaceutics13101588. PMID: 34683881; PMCID: PMC8538955.
  • Hsu-Cheng Chiang, Bailey Eberle, Drew Carlton, Thomas J. Kolibaba, and Jaime C. Grunlan. Edible Polyelectrolyte Complex Nanocoating for Protection of Perishable Produce. ACS Food Science & Technology 2021 1 (4), 495-499 DOI: 10.1021/acsfoodscitech.1c00097

Comment 2: Line 81. The authors mention that ginger oil contains phenolic substances, which are responsible for the necessary properties of the resulting material, but Figure 1 shows quite different substances and no phenolic substances. There is an additional question to this same figure, which is not mentioned by the authors in the article. The point is that the terpenoids depicted are volatile and have a specific odor, which can affect the product packaged in the material. Indeed, if we are talking about the packaging of gingerbread, then the presence of volatile ginger fragrances in the packaging will only benefit them, but if we are talking about products that could be harmed by such flavoring, it is at least worth discussing.

Response: Thanks for your valuable suggestion. Figure 1 is modified as per your valuable suggestion. I agreed with your valuable suggestion that addition of any aromatic or favouring components may interfere with original sensory properties of the packed food material. However, leaching of essential components from the film to the packed food material for relatively shorter duration would be having less negative impact than leaching of harmful components from the plastic material for longer duration. Also, less concertation, non-polar and volatile nature wouldn’t allow its entry inside to the food material (only on surface). Thus, it will only act on the surface.  Additionally, it contributes to antioxidant and antimicrobial effects. USFDA also considered GEO in the category of GRAS (generally recognized as safe, plz refer the following article). Also, there are several reports suggested that use of essential oil in food preservation is beneficial.

Sharifi-Rad M, Varoni EM, Salehi B, Sharifi-Rad J, Matthews KR, Ayatollahi SA, Kobarfard F, Ibrahim SA, Mnayer D, Zakaria ZA, Sharifi-Rad M, Yousaf Z, Iriti M, Basile A, Rigano D. Plants of the Genus Zingiber as a Source of Bioactive Phytochemicals: From Tradition to Pharmacy. Molecules. 2017 Dec 4;22(12):2145. doi: 10.3390/molecules22122145. PMID: 29207520; PMCID: PMC6149881.

Comment 3: Lines 478-479, 485-486. The description of the materials used does not allow us to reproduce the study described, nor does it allow us to get an idea of what materials were used. Since ginger oil, chitosan, and porphyran are materials of natural origin, they are characterized by a high variability in composition and structure, molecular weights and their distribution, etc. For ginger oil, it would be useful to give at least the data of chromato-mass spectrometric analysis; for polysaccharides, a description of the original source of the materials and gel-exclusion chromatograms are necessary so that the reader can imagine exactly what materials were used. Indeed, for the practical use of polysaccharides, one of the main obstacles is the poor reproducibility in their preparation, insufficiently adequate analysis of the synthesized material, which often makes it impossible to reproduce the result obtained.

Response: Thanks for your valuable suggestion. Since the Ginger oil in pure form was purchased from the Natures Natural India accredited by ISO 9001-2015, US FDA, WHO, GMP, HACCP and HALAL for their quality standards. Company has also provided the certificate of analysis of all oils which have been purchased (Zingiberene content 11.5 %; Color and Appearance Pale yellow to brown and mobile liquid Odor Aromatic, ginger Refractive Index 1.481 – 1.499 Specific Gravity 0.896 – 0.912 Optical Rotation -28 Flash Point 99°C).  Chitosan (extra pure, 150-500m.Pas, 90% DA) was purchased from SRL Pvt. Ltd. India. Isolation and characterization of porphyran is already mentioned in our previous papers (listed below) and cited in the present work also:

  • Bhatia S, Kumar V, Sharma K, Nagpal K, Bera T. Significance of algal polymer in designing amphotericin B nanoparticles. ScientificWorldJournal. 2014;2014:564573.
  • Bhatia S, Rathee P, Sharma K, Chaugule BB, Kar N, Bera T. Immuno-modulation effect of sulphated polysaccharide (porphyran) from Porphyra vietnamensis. Int J Biol Macromol. 2013 Jun;57:50-6. doi: 10.1016/j.ijbiomac.2013.03.012. Epub 2013 Mar 13. PMID: 23500431.
  • Saurabh Bhatia, S. Bhatia, Kiran Sharma, K. Sharma, Kalpana Nagpal, K. Nagpal, & Tanmoy Bera, T. Bera. (0000). Investigation of the factors influencing the molecular weight of porphyran and its associated antifungal activity. Bioactive carbohydrates and dietary fibre, 5, 153-168.
  • Bhatia, M. S. (2015). Structural characterization and pharmaceutical properties of porphyran. Asian Journal of Pharmaceutics (AJP): Free Full Text Articles from Asian J Pharm, 9(2), 93–101.

Regarding the reproducibility of the results using the natural polymers and oils, yes, I agreed that reproducibility of the result may affect when natural polymers and essential oil or any other natural component used however primary objective of the research article is to focus on the development, characterization and biological assessment of the edible films.  It would have been interesting to explore this aspect as well. Based on your valuable suggestion we will certainly plan characterization of the polymer and oils soon. Regarding isolation and characterization of porphyran its already done in our previous articles listed above.  Also we have followed the following and many other papers where  based on their primary objectives they have excluded this part (characterization of essential oil and polymers).

  • Tan LF, Elaine E, Pui LP, Nyam KL, Yusof YA. Development of chitosan edible film incorporated with Chrysanthemum morifolium essential oil. Acta Sci Pol Technol Aliment. 2021 Jan-Mar;20(1):55-66. doi: 10.17306/J.AFS.0771. PMID: 33449520.
  • Khazaei A, Nateghi L, Zand N, Oromiehie A, Garavand F. Evaluation of Physical, Mechanical and Antibacterial Properties of Pinto Bean Starch-Polyvinyl Alcohol Biodegradable Films Reinforced with Cinnamon Essential Oil. Polymers (Basel). 2021 Aug 18;13(16):2778. doi: 10.3390/polym13162778. PMID: 34451316; PMCID: PMC8399529.
  • Hasheminya SM, Dehghannya J. Development and characterization of novel edible films based on Cordia dichotoma gum incorporated with Salvia mirzayanii essential oil nanoemulsion. Carbohydr Polym. 2021 Apr 1;257:117606. doi: 10.1016/j.carbpol.2020.117606. Epub 2021 Jan 4. PMID: 33541639.
  • Mahcene Z, Khelil A, Hasni S, Akman PK, Bozkurt F, Birech K, Goudjil MB, Tornuk F. Development and characterization of sodium alginate based active edible films incorporated with essential oils of some medicinal plants. Int J Biol Macromol. 2020 Feb 15;145:124-132. doi: 10.1016/j.ijbiomac.2019.12.093. Epub 2019 Dec 13. PMID: 31843601.
  • Simona J, Dani D, Petr S, Marcela N, Jakub T, Bohuslava T. Edible Films from Carrageenan/Orange Essential Oil/Trehalose-Structure, Optical Properties, and Antimicrobial Activity. Polymers (Basel). 2021 Jan 21;13(3):332. doi: 10.3390/polym13030332. PMID: 33494246; PMCID: PMC7864528.
  • Zhou Y, Wu X, Chen J, He J. Effects of cinnamon essential oil on the physical, mechanical, structural and thermal properties of cassava starch-based edible films. Int J Biol Macromol. 2021 Aug 1;184:574-583. doi: 10.1016/j.ijbiomac.2021.06.067. Epub 2021 Jun 16. PMID: 34146564.
  • Arezoo E, Mohammadreza E, Maryam M, Abdorreza MN. The synergistic effects of cinnamon essential oil and nano TiO2 on antimicrobial and functional properties of sago starch films. Int J Biol Macromol. 2020 Aug 15;157:743-751. doi: 10.1016/j.ijbiomac.2019.11.244. Epub 2019 Dec 2. PMID: 31805325.
  • Guo Y, Chen X, Yang F, Wang T, Ni M, Chen Y, Yang F, Huang D, Fu C, Wang S. Preparation and Characterization of Chitosan-Based Ternary Blend Edible Films with Efficient Antimicrobial Activities for Food Packaging Applications. J Food Sci. 2019 Jun;84(6):1411-1419. doi: 10.1111/1750-3841.14650. Epub 2019 May 27. PMID: 31132162.
  • Chu Y, Xu T, Gao C, Liu X, Zhang N, Feng X, Liu X, Shen X, Tang X. Evaluations of physicochemical and biological properties of pullulan-based films incorporated with cinnamon essential oil and Tween 80. Int J Biol Macromol. 2019 Feb 1;122:388-394. doi: 10.1016/j.ijbiomac.2018.10.194. Epub 2018 Oct 30. PMID: 30385340.

In addition to the above comments, all spelling and grammatical errors pointed out by the reviewers have been corrected. We look forward to hearing from you in due time regarding our submission and to respond to any further questions and comments you may have.

Sincerely

Round 2

Reviewer 3 Report

Comment 1: Lines 68-69.  If the authors insist on using the term complex instead of salt, then I do not insist on replacing the word complex with salt.

Comment 2: Line 81. The authors mention that «However, leaching of essential components from the film to the packed food material for relatively shorter duration would be having less negative impact than leaching of harmful components from the plastic material for longer duration. Also, less concertation, non-polar and volatile nature wouldn’t allow its entry inside to the food material (only on surface). Thus, it will only act on the surface.  Additionally, it contributes to antioxidant and antimicrobial effects. USFDA also considered GEO in the category of GRAS (generally recognized as safe, plz refer the following article). Also, there are several reports suggested that use of essential oil in food preservation is beneficial.» Perhaps they should add this reasoning to the article?

Comment 3: Lines 478-479, 485-486. «Since the Ginger oil in pure form was purchased from the Natures Natural India accredited by ISO 9001-2015, US FDA, WHO, GMP, HACCP and HALAL for their quality standards. Company has also provided the certificate of analysis of all oils which have been purchased (Zingiberene content 11.5 %; Color and Appearance Pale yellow to brown and mobile liquid Odor Aromatic, ginger Refractive Index 1.481 – 1.499 Specific Gravity 0.896 – 0.912 Optical Rotation -28 Flash Point 99°C). » Perhaps the authors should add this data to the article?

«Chitosan (extra pure, 150-500m.Pas, 90% DA) was purchased from SRL Pvt. Ltd. India. Isolation and characterization of porphyran is already mentioned in our previous papers (listed below) and cited in the present work also:» I must regret to state that it is the poor characterization and standardization of chitosans that often inhibits or even cancels their use in practice. Indeed, one production facility sometimes manages to produce some product of more or less the same quality, but if we want to reproduce it elsewhere, it usually fails for unknown reasons. And if there is a description of the experimental procedure for the porphyran obtained by the authors, there is no such data about the chitosan purchased. What is the source of this chitosan (insects, larvae, crustaceans, fungi)? The molecular weight distribution, the presence of defects in the polymer structure, the presence of impurities in the polymer (which is especially critical for chitosans of mushroom origin) dramatically affect the properties of chitosan materials. If the manufacturer does not provide chromatographic data for his product, why don't the authors do gel-exclusion chromatography? This would enable the reader, if not to understand the structure of a given chitosan, then at least, having bought chitosan in the same place, to compare the starting material by the "fingerprint" method. As a rule, producers working on natural rather than cultivated raw materials ignore biological type of crustaceans (they can use different types of shrimps or crabs from synthesis to synthesis), maturity of raw materials, i.e. age of organisms, etc. The chitosan obtained in this way differs significantly in the characteristics that are important for making materials from them, not just gels.

« Based on your valuable suggestion we will certainly plan characterization of the polymer and oils soon.» I understand and agree that the porphyran obtained by the authors can be characterized in the following works, as well as more or less "standard" ginger oil. The former is the object of their own research and is isolated from the same species of algae, the latter is a commercial food product whose "taste" is closely monitored by tasters. But the situation with the chitosan purchased by the authors is different from these cases.

Author Response

Dear Reviewer,

Thank you for giving me the opportunity to submit a revised draft of my manuscript titled “Development and Characterization of Chitosan and Porphyran Based Composite Edible Films Containing Ginger Essential Oil” to Polymers, MDPI. I appreciate the time and effort that you have dedicated to providing your valuable feedback on my manuscript. I am grateful to the reviewers for their insightful comments on my paper. I have been able to incorporate changes to reflect most of the suggestions provided by the reviewers. I have highlighted the changes within the manuscript and submit it accordingly. Here is a point-by-point response to the reviewers’ comments and concerns.

Comment 1: Lines 68-69.  If the authors insist on using the term complex instead of salt, then I do not insist on replacing the word complex with salt.

Thank you so much for allowing us to retain word “complex”. In the manuscript we have used the word complex now. 

Comment 2: Line 81. The authors mention that «However, leaching of essential components from the film to the packed food material for relatively shorter duration would be having less negative impact than leaching of harmful components from the plastic material for longer duration. Also, less concertation, non-polar and volatile nature wouldn’t allow its entry inside to the food material (only on surface). Thus, it will only act on the surface.  Additionally, it contributes to antioxidant and antimicrobial effects. USFDA also considered GEO in the category of GRAS (generally recognized as safe, plz refer the following article). Also, there are several reports suggested that use of essential oil in food preservation is beneficial. Perhaps they should add this reasoning to the article?

Thank you so much for this valuable suggestion. We have included this information and highlighted also.

Comment 3: Lines 478-479, 485-486. «Since the Ginger oil in pure form was purchased from the Natures Natural India accredited by ISO 9001-2015, US FDA, WHO, GMP, HACCP and HALAL for their quality standards. Company has also provided the certificate of analysis of all oils which have been purchased (Zingiberene content 11.5 %; Color and Appearance Pale yellow to brown and mobile liquid Odor Aromatic, ginger Refractive Index 1.481 – 1.499 Specific Gravity 0.896 – 0.912 Optical Rotation -28 Flash Point 99°C). » Perhaps the authors should add this data to the article?

Thank you so much for this valuable suggestion. We have included this information and highlighted also.

«Chitosan (extra pure, 150-500m.Pas, 90% DA) was purchased from SRL Pvt. Ltd. India. Isolation and characterization of porphyran is already mentioned in our previous papers (listed below) and cited in the present work also: I must regret to state that it is the poor characterization and standardization of chitosans that often inhibits or even cancels their use in practice. Indeed, one production facility sometimes manages to produce some product of more or less the same quality, but if we want to reproduce it elsewhere, it usually fails for unknown reasons. And if there is a description of the experimental procedure for the porphyran obtained by the authors, there is no such data about the chitosan purchased. What is the source of this chitosan (insects, larvae, crustaceans, fungi)? The molecular weight distribution, the presence of defects in the polymer structure, the presence of impurities in the polymer (which is especially critical for chitosans of mushroom origin) dramatically affect the properties of chitosan materials. If the manufacturer does not provide chromatographic data for his product, why don't the authors do gel-exclusion chromatography? This would enable the reader, if not to understand the structure of a given chitosan, then at least, having bought chitosan in the same place, to compare the starting material by the "fingerprint" method. As a rule, producers working on natural rather than cultivated raw materials ignore biological type of crustaceans (they can use different types of shrimps or crabs from synthesis to synthesis), maturity of raw materials, i.e. age of organisms, etc. The chitosan obtained in this way differs significantly in the characteristics that are important for making materials from them, not just gels.

Based on your valuable suggestion we will certainly plan characterization of the polymer and oils soon.» I understand and agree that the porphyran obtained by the authors can be characterized in the following works, as well as more or less "standard" ginger oil. The former is the object of their own research and is isolated from the same species of algae, the latter is a commercial food product whose "taste" is closely monitored by tasters. But the situation with the chitosan purchased by the authors is different from these cases.

Thanks for your valuable suggestion. We have already included chitosan source and its specification [Chitosan (extrapure, 150-500m.Pas, 90% DA by SRL Pvt. Ltd. India)], and now we have highlighted for your convenience. As in majority of the previous report’s chitosan (90% DA) was purchased from manufacturer/ distributor and no one has performed this analysis (chromatography), so we planned in the same way. As per manufacturer policies I am not sure whether the manufacturer will disclose the source and process for the production of the chitosan. I agreed with your comment that each company may use different source or material to produce chitosan, however the most important factor is percentage of deacetylation which can determine the various properties of the polymer significantly. As per your valuable suggestion we will certainly plan to study the effect of different grades of (degree of deacetylation 50-90) over the physical chemical properties of the film.

Thank you so much for sparing your time for our work and giving us valuable feedback. We look forward to hearing from you in due time regarding our submission and to respond to any further questions and comments you may have.

Sincerely
